# Association of Circulating miR-145-5p and miR-let7c and Atherosclerotic Plaques in Hypertensive Patients

**DOI:** 10.3390/biom11121840

**Published:** 2021-12-07

**Authors:** Eduarda O. Z. Minin, Layde R. Paim, Elisangela C. P. Lopes, Larissa C. M. Bueno, Luís F. R. S. Carvalho-Romano, Edmilson R. Marques, Camila F. L. Vegian, José A. Pio-Magalhães, Otavio R. Coelho-Filho, Andrei C. Sposito, José R. Matos-Souza, Wilson Nadruz, Roberto Schreiber

**Affiliations:** Department of Internal Medicine, School of Medical Sciences, State University of Campinas, Campinas 13083-970, São Paulo, Brazil; dudaminin@gmail.com (E.O.Z.M.); layde_rosane@yahoo.com.br (L.R.P.); elisangela.cpl@gmail.com (E.C.P.L.); larissacmbueno@gmail.com (L.C.M.B.); luisfelipersc@gmail.com (L.F.R.S.C.-R.); edrmarques@yahoo.com.br (E.R.M.); cvegian@unicamp.br (C.F.L.V.); alexandre@hc.unicamp.br (J.A.P.-M.); tavicocoelho@gmail.com (O.R.C.-F.); andreisposito@gmail.com (A.C.S.); betojrms@gmail.com (J.R.M.-S.); wilnj@unicamp.br (W.N.)

**Keywords:** hypertension, carotid atherosclerosis, miRNAs

## Abstract

Aim: Hypertension is a strong risk factor for atherosclerosis. Increased carotid intima-media thickness (cIMT) and carotid plaques are considered subclinical markers of atherosclerosis. This study aimed at evaluating the serum expression of miRNAs previously related to adverse vascular remodeling and correlating them with carotid plaques and cIMT in hypertensive patients. Methods: We cross-sectionally evaluated the clinical and carotid characteristics as well as serum expression of miR-145-5p, miR-let7c, miR-92a, miR-30a and miR-451 in 177 hypertensive patients. Carotid plaques and cIMT were evaluated by ultrasound, and the expression of selected miRNAs was evaluated by a quantitative polymerase chain reaction. Results: Among all participants (age = 60.6 ± 10.7 years, 43% males), there were 59% with carotid plaques. We observed an increased expression of miR-145-5p (Fold Change = 2.0, *p* = 0.035) and miR-let7c (Fold Change = 3.8, *p* = 0.045) in participants with atherosclerotic plaque when compared to those without plaque. In the logistic regression analysis adjusted for relevant covariates, these miRNAs showed a stronger association with carotid plaques (miR-145-5p: *Beta ± SE* = 0.050 ± 0.020, *p* = 0.016 and miR-let7c: *Beta ± SE* = 0.056 ± 0.019, *p* = 0.003). Conclusions: Hypertensive patients with carotid plaques have an increased expression of miR-145-5p and miR-let7c, suggesting a potential role of these miRNAs as a biomarker for subclinical atherosclerosis in hypertensive individuals.

## 1. Introduction

Atherosclerotic cardiovascular diseases are the main cause of mortality worldwide [1]. In the primary prevention setting, the presence of carotid atherosclerotic plaques and increased carotid intima media thickness (cIMT) are considered in the definition of subclinical atherosclerosis and, as such, are predictors of cardiovascular events [2]. Hypertension is a major risk factor for atherosclerosis in distinct population grounds [3,4]. In line with this, high blood pressure (BP) levels are associated with increased cIMT and the presence of carotid plaques [5,6,7].

MicroRNAs (miRNAs) are a class of small noncoding RNAs that normally inhibit translation, resulting in the degradation of the target mRNA. Currently, more than 2600 miRNAs are known in humans, which have been involved in both physiological and pathological processes [8,9]. Available evidence suggests that miRNAs might play an important role in atherogenesis and vascular remodeling by acting as a regulator of cell growth and proliferation and of inflammation [10]. In this regard, several miRNAs, including miR-145-5p, miR-let7c, miR-92a, miR-30a and miR-451, have been reported to be associated with atherosclerosis and/or adverse vascular remodeling [11,12,13,14,15]. However, the importance of these miRNAs as markers of atherosclerotic burden among hypertensive patients is uncertain.

The present study aimed at evaluating the serum expression of miR-145-5p, miR-let7c, miR-92a, miR-30a and miR-451, and correlating them with carotid plaques and cIMT in hypertensive patients.

## 2. Materials and Methods

### 2.1. Study Population

In this study, the clinical and echocardiographic characteristics of 177 consecutive hypertensive patients followed at a university outpatient clinic were evaluated. Exclusion criteria included age < 18 years, hypertrophic cardiomyopathy, and moderate or severe valve disease. The study was approved by the Ethics Committee of the State University of Campinas and follows the guidelines of the Declaration of Helsinki of 1975 (CAAE:56841616.5.0000.5404). Written consent was obtained from all participants.

### 2.2. Clinical Characteristics

The blood pressure and heart rate were measured prior to echocardiographic analysis using a validated digital oscillometer device (HEM-7113, Omron Corp., Kyoto, Japan) with appropriate cuff sizes. The mean of two BP readings was calculated, and when they differed by more than 5 mmHg a new reading was taken, and then the mean of the three measurements was calculated. Hypertension was defined as office BP 140/90 mmHg or the use of antihypertensive medications. The pulse pressure was calculated as Systolic BP–Diastolic BP. The body mass index (BMI) was calculated as the weight divided by height squared (kg/m^2^). Fasting blood glucose, creatinine, triglycerides, low-density lipoprotein cholesterol (LDL-C) and high-density lipoprotein cholesterol (HDL-C), and glycated hemoglobin were measured using standard laboratory techniques. Hypercholesterolemia was defined as LDL-C > 130 mg/dL or the use of cholesterol-lowering medications. Type 2 diabetes mellitus (T2DM) was diagnosed as fasting glucose ≥ 126 mg/dL or glycated hemoglobin ≥ 6.5 g/dL or the use of antidiabetic medication.

### 2.3. Carotid Ultrasound

High resolution images of the right and left common carotid arteries (CCA) were obtained 2 cm proximal to each participant’s carotid bifurcation. A Vivid q device (General Electric, Milwaukee, WI, USA) equipped with a linear transducer (12L-RS; 6–13 MHz) set at 10 MHz was used, as previously reported [16,17], and analyses were performed by an experienced physician. For each image, five measurements of far-wall cIMT were manually performed by a physician (L.F.R.S.C–R.) in plaque-free areas using the ImageJ software (NIH, Bethesda, Maryland, USA), and the average of the left and right CCA measurements was used to calculate the thickness [16,17]. Carotid plaques were defined as CCA cIMT ≥1.5 mm or focal wall thickening encroaching into the lumen by 50% or 0.5 mm. The intra observer and inter observer variability were 1.2% and 3.5%, respectively, when considering the analysis of 20 images from 10 patients.

### 2.4. RNA Isolation and Quantitative Real Time PCR (qRT-PCR)

The miRNA was extracted from serum samples using the miRNeasy Serum/Plasma Kit (Qiagen, Valencia, CA, USA). The quantity and quality of miRNAs were measured by NanoDrop ND-2000 Spectrophotometer (Thermo Fisher Scientific, Waltham, MA, USA) as previously reported [18]. The reverse transcription (RT) reactions were run in a Mastercycler ep (Eppendorf, Hamburg, Germany) 96-Well Thermal Cycler according to the manufacturer’s instructions and performed using the SuperScript^®^III First Strand Synthesis Kit (Applied Biosystems, Waltham, MA, USA).

The expression levels of miRNAs were run in triplicate and detected by the miRNA qRT-PCR Assay Kit for miR-145-5p (002278), miR-let7c (000379), miR-92a (000431), miR-30a (000417) and miR-451 (001141) in StepOne Plus Real-Time PCR Systems (Thermo Fisher Scientific, Inc.). The reactions were heated to 95 °C for 10 min, followed by 40 cycles of 95 °C for 15 s and 60 °C for 1 min. The relative expression of miRNA was calculated with the comparative threshold cycle (2^−^^ΔΔCt^) method, and the fold change (FC) was calculated as FC = 2^−ΔΔCt^, where Ct is defined as the PCR cycle number at which the fluorescence meets the threshold in the amplification plot [19]. Data were normalized using a geometric mean of U6 snRNA (noncoding small nuclear RNA-001973) and miR-16 (000391) as the housekeeping genes.

### 2.5. Gene Set Enrichment Analysis

To understand the biological relevance of differentially expressed miRNAs, we performed a functional enrichment analysis. The differentially expressed miRNAs that correlated between patients with and without carotid plaque were loaded into miRWalk 2.0 [20]. Only mRNAs predicted in at least four of the five tools (miRanda, miRDB, miRWalk, RNA22 and TargetScan) were considered as possible miRNA targets. We used the Database for Integrated Annotation, Visualization and Discovery (DAVID) to obtain Gene Ontology (GO) and Kyoto Encyclopedia of Genes and Genomes (KEGG) to determine the enriched pathways. In addition, we used Cytoscape software [21] to build a network displaying miRNAs and their gene targets.

### 2.6. Statistical Analysis

Statistical analyses were performed using SPSS software (SPSS 16.0). Variables with a normal or non-normal distribution are presented as the mean ± standard deviation (SD) and median [25th, 75th percentiles], and their differences between individuals with and without plaques were evaluated by an unpaired student’s t-test and Mann–Whitney U-test, respectively. A chi-square test was used to compare categorical variables. The correlation between cIMT and the expression of miRNAs was assessed by the Pearson’s method. The evaluation of the association between carotid plaques or cIMT and the expression of miRNAs was assessed by a multivariable linear regression analysis adjusted for age, sex, T2DM, BMI, pulse pressure, LDL-C antihypertensive medications and statins. A *p*-value < 0.05 was considered statistically significant.

## 3. Results

### 3.1. Clinical Characteristics of Participants

Among all participants (age = 60.6 ± 10.7 years, 43% males), the mean cIMT was 0.746 ± 0.140, and 59% presented carotid plaques. The clinical, laboratory and cIMT measures of the participants according to the presence or not of atherosclerotic plaque are presented in Table 1. Participants with plaques were older, had a lower diastolic BP, and higher values of creatinine and cIMT than those without plaques.

### 3.2. Relationship between miRNA and Carotid Plaques and cIMT

qRT-PCR was used to analyze the level of miRNA expression in serum samples from hypertensive patients. The results showed that the expression level of miR-145-5p and miR-let7c in the serum of patients with carotid plaques was higher than in those without plaques (FC = 2, *p* = 0.035; FC = 3.8, *p* = 0.045), respectively (Figure 1).

The results of the multivariable linear regression analysis showed an association of plaque presence with miR-145-5p (*Beta ± SE* = 0.047 ± 0.020; *p* = 0.022) and miR-let7c (*Beta ± SE* = 0.056 ± 0.018; *p* = 0.003) levels after adjustment for age, sex, pulse pressure, BMI, T2DM, LDL-C, smoking, statins use and therapy with antihypertensive drugs.

The results of the bivariate correlation analysis showed a correlation of the serum expression of miR-145-5p with cIMT (*r* = 0.195; *p* = 0.013) in hypertensive patients (Figure 2), which remained significant after adjustment for confounding variables such as age, sex, pulse pressure, BMI, T2DM, LDL-C, smoking, statins use and antihypertensive therapy in the multivariate linear regression analysis (*Beta ± SE* = 0.016 ± 0.007; *p* = 0.022). No further correlation between other studied miRNAs and cIMT was found.

Using miRWALK2.0 software, 1644 genes were identified as potential genes targeted by miR-145-5p and miR-let7c (Appendix A), with 111 being common to the two miRNAs (Figure 3). The GO results showed that regarding the biological process, the term “response to cholesterol” was the most enriched, followed by the terms related to “Calcium and WNT signaling pathways”, while in the cellular components, the term “alphaV-beta3 integrin-IGF-1-IGF1R complex” was the most enriched. In the molecular functions, the terms “Wnt-activated receptor activity” and “Wnt-protein binding” were the most enriched. The results of the KEGG enrichment analysis showed that some pathways were related to atherosclerosis such as the Hippo, FoxO signaling pathways and the term “Adherens junction” (Figure 4).

## 4. Discussion

In the present study, we measured the serum expression of five miRNAs previously described as playing a potential role in vascular remodeling (miR-145-5p, miR-let7c, miR-92a, miR-30a and miR-451) and evaluated their association with carotid atherosclerotic plaques and cIMT in a sample of hypertensive patients. We observed increased circulating levels of miR-145-5p and miR-let7c expression in hypertensive patients with atherosclerotic plaque, even adjusting for relevant confounders. Conversely, only miR-145-5p had a significant association with cIMT. Because the presence of carotid plaques is considered a more specific marker of atherosclerosis than cIMT [22], the present findings suggest that both miR-145-5p and miR-let7c might be involved in atherogenesis among hypertensive subjects.

Reports regarding the role of miR-145 in atherosclerosis have yielded conflicting results. Previous data obtained in experimental models suggested an atheroprotective role of miR-145 [23,24], especially by modulating the switch of vascular smooth muscle cells from a proliferative and migratory to a contractile state [23].

By contrast, longitudinal studies have shown that an increased expression of miR-145 may be related to the development of atherosclerosis [11,25]. For instance, Santovito et al. [11] demonstrated an increased expression of miR-145 in carotid atherosclerotic plaques of hypertensive patients when compared to plaques from individuals without hypertension, while Knoka et al. [25] demonstrated an association between higher circulating levels of miR-145 and the vulnerability of coronary plaques. Our study, in agreement with both studies, reported that higher circulating levels of miR-145-5p were associated with carotid plaques in hypertensive patients. The reason for the apparent discrepancies between the results of the experimental and clinical studies are not clear, but some explanations may be suggested. First, increases in mir-145 expression may be part of a negative feedback mechanism that impedes plaque progression, as suggested by Knoka et al. [25]. In this case, the increase in miR-145 expression in the vulnerable plaque would prevent its further destabilization. However, we do not have data on miRNA expression in atherosclerotic plaque to support this hypothesis. Second, it is possible that the role of miR-145 in atherogenesis may vary depending on the disease stage. Although increased miR-145 expression may have an atheroprotective effect in the early stages of atherosclerosis, it may contribute to the development of plaque destabilization in later stages [25].

T cells, dendritic cells and macrophages, in addition to oxidized LDL, are main components of atherosclerotic plaques [26]. In two studies, Frostegards et al. demonstrated that miR-let7c expression was increased in atherosclerotic plaque induced by increases in oxidized LDL, an effect that can be abolished by statins [27] and to a lesser degree by PSCK9 inhibition [28]. Furthermore, miR-let7c was reported to play an important role in the activation of T and dendritic cells induced by oxidized LDL [27]. In another study, Huang et al. observed a higher plasma expression of miR-let7c in hypertensive patients with cIMT and a positive correlation between miRNA and cIMT confirmed by a multiple linear regression analysis [12]. Overall, these data suggest that miR-let7c may exert pro-atherogenic effects in both experimental and clinical settings. In agreement with this assumption, we found an increased expression of miR-let7c in the serum of hypertensive patients with atherosclerotic plaques when compared to hypertensive patients without apparent atherosclerotic plaques. Interestingly, we did not observe a correlation between miR-let7c and cIMT. In this regard, it is important to recognize that the presence of carotid atherosclerotic plaques, as a later manifestation of atherosclerotic disease, is a stronger predictor of cardiovascular events as compared with cIMT [29]. Furthermore, this may also be due to the fact that cIMT measures the carotid intima and medial wall layers [22]. In this sense, the association we found between miR-let7c and atherosclerotic plaque provides more robust evidence that this miRNA might be related to atherosclerosis.

A functional enrichment analysis was performed to identify their target genes and pathways to understand the relevance of these two miRNAs, and we observed that miR-145-5p and miR-let7c regulated potential genes and pathways related to metabolic signaling pathways, as well as inflammatory, focal and calcium ones. In particular, our in-silico analysis detected at least nine target genes, ADRB3 [30], CBL [31], IGF1 and IGF1R [32], SMAD2 and TGFBR1 [33], NFIA [34], TRIM13 and TRIM65 [35], with reports in the literature of a reduced expression in the atherosclerotic plaque, suggesting that these genes may be involved in the atherogenesis process regulated by miR-145-5p and let7c.

The analysis of KEGG revealed the involvement of signaling pathways associated with atherosclerosis, such as Hippo [36], FoxO [37] and Adherens junction [38], while the gene ontology results indicated for both “Biological Processes” and “Molecular Function” the participation of genes associated with the WNT signaling pathway, which has been linked to the pathophysiology of atherosclerosis [39].

We recognize that our study has some limitations. We did not evaluate the expression of these miRNAs in atherosclerotic plaque, but our findings appear to reproduce in serum the results from other studies evaluating the expression of miRNA in plaques of hypertensive patients [11]. As with any cross-sectional study, the influence of residual confounding cannot be excluded, and the association between these miRNAs and the presence of plaque must be carefully evaluated. Furthermore, it is possible that the use of medications may have influenced our findings. Notably, miR-145 expression may be influenced by some antihypertensive drugs, like angiotensin-converting enzyme inhibitors and angiotensin receptor blockers [40]. To overcome this limitation, we included potential confounding variables and the use of each class of antihypertensive medication and statins as independent variables in our multivariable analyses.

## 5. Conclusions

In summary (Appendix A), our data demonstrate that hypertensive patients with carotid plaque have an increased expression of miR-145-5p and miR-let7c, suggesting that these miRNAs can be used as potential biomarkers of carotid atherosclerosis in patients with arterial hypertension.

## Figures and Tables

**Figure 1 biomolecules-11-01840-f001:**
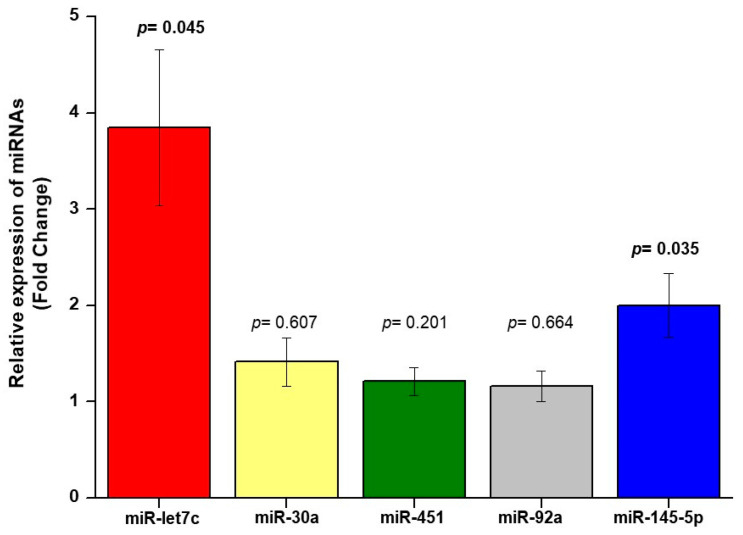
Differential expression analysis of miRNAs obtained in the serum of hypertensive patients. Differentially expressed serum miRNAs (fold change) in patients with plaque compared with patients without plaque. *p*-value from independent Mann–Whitney test is presented.

**Figure 2 biomolecules-11-01840-f002:**
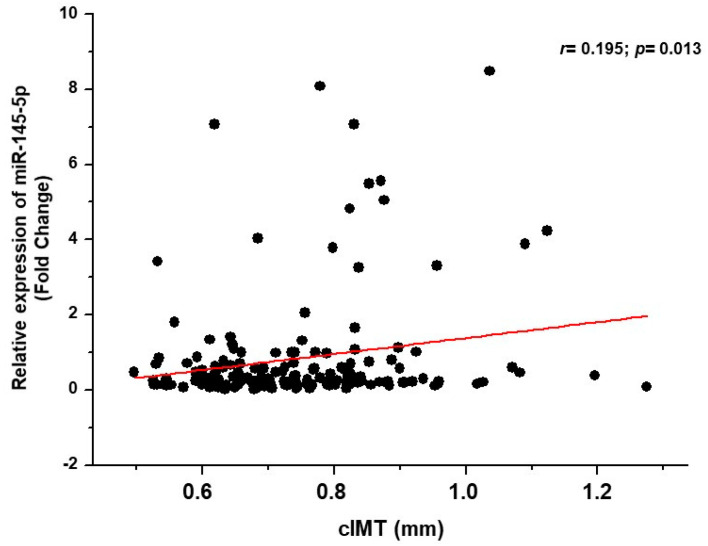
Bivariate correlation analysis between the serum expression of miR-145-5p and cIMT in hypertensive patients.

**Figure 3 biomolecules-11-01840-f003:**
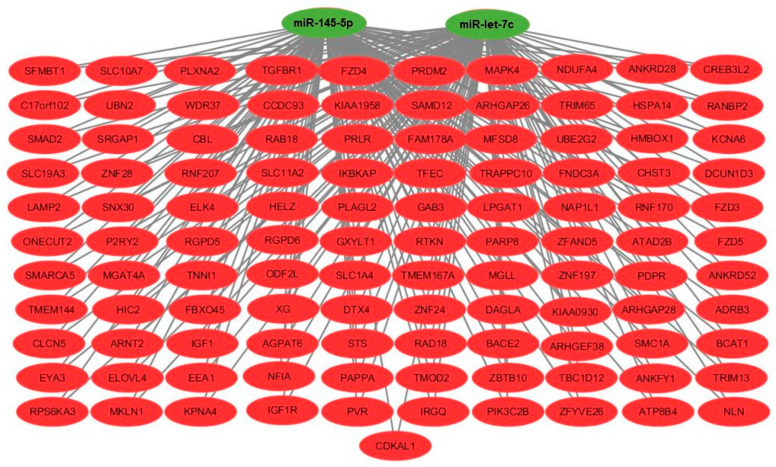
Network model representing the potential target genes of each differentially regulated miRNA in our study. The downregulated ones are in red, the upregulated ones are in green.

**Figure 4 biomolecules-11-01840-f004:**
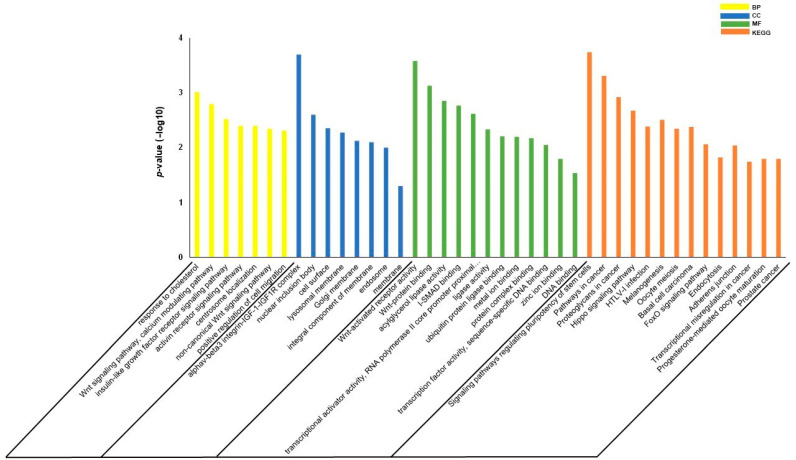
Pathway enrichment analysis of predicted target genes. Enrichment analysis of target genes common to the two miRNAs (miR-145-5p and miR-let7c) that correlated with the presence of plaque in hypertensive patients. Biological process (BP) terms are shown as yellow bars, cellular component (CC) terms are shown as blue bars, molecular function (MF) terms are shown as green bars, and KEGG pathways are shown as orange bars according to the *p*-value (<0.05) −log10 transformed.

**Table 1 biomolecules-11-01840-t001:** Characteristics of the participants according to the presence of carotid plaques.

Variables	Plaques–No	Plaques–Yes	*p*-Value
	*n* = 72 (40%)	*n* = 105 (59%)	
Clinical			
Age, years	55.0 ± 11.2	64.3 ± 8.6	<0.001
Male sex, %	42	45	0.80
Systolic blood pressure, mmHg	150.7 ± 25.7	152.4 ± 25.6	0.67
Diastolic blood pressure, mmHg	88.1 ± 16.4	81.5 ± 14.9	0.007
Pulse pressure, mmHg	62.7 ± 17.3	70.7 ± 20.3	0.006
Body mass index, kg/m^2^	30.3 ± 6.1	29.9 ± 5.4	0.68
Diabetes mellitus, %	48	57	0.43
Current smoking, %	7	7	0.96
Coronary artery disease, %	19	27	0.79
Previous stroke, %	12	19	0.74
Diuretics, %	71	69	0.87
ACEI or ARB, %	85	80	0.55
Beta-blocker, %	58	63	0.65
Calcium-channel blocker, %	64	59	0.62
Statin, %	65	74	0.26
Glucose, mg/dL	99 [87, 108]	103 [92, 133]	0.07
Triglycerides, mg/dL	124 [90, 173]	154 [103, 204]	0.26
HDL-C, mg/dL	46.4 ± 13.4	44.1 ± 12.5	0.25
LDL-C, mg/dL	96.9 ± 28.5	90.4 ± 35.5	0.19
Creatinine, mg/dL	0.9 [0.7, 1.1]	1.1 [0.8, 1.3]	0.039
cIMT, mm	0.669 ± 0.103	0.799 ± 0.138	<0.001

Continuous data with normal and non-normal distribution are presented as mean ± standard deviation and median [25th, 75th percentiles]. ACEI or ARB—angiotensin-converting enzyme inhibitor or angiotensin receptor blocker; HDL-C—high density lipoprotein-cholesterol; LDL-C—low density lipoprotein-cholesterol; cIMT—carotid intima-media thickness.

## Data Availability

All data presented in this study are available upon request for correspondence from the author, and the raw data will be archived in an Institutional Data Repository.

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
