# Peer review of "Association of Circulating miR-145-5p and miR-let7c and Atherosclerotic Plaques in Hypertensive Patients"

_biomolecules, 2021, doi:10.3390/biom11121840_

Round 1
Reviewer 1 Report
Dear Authors,
The manuscript entitled: “Association of circulating miR-145-5p and miR-let7c and atherosclerotic plaques in hypertensive patients.” Is very interesting and it brings new information about the involvement miR-145-5p and miR-let7c in hypertensive patients with or without the atherosclerotic plaques. I found the publication well written, the conclusions are clear and supported by the results.
I have one comment, in the conclusion is written: “In summary, our data demonstrate that hypertensive patients with carotid plaque have increased expression of miR-145-5p and miR-let7c, suggesting a potential role as biomarkers of carotid atherosclerosis in patients with arterial hypertension.” Maybe the sentence should be modified to suggest that those miRNA might be used as biomarkers in hypertensive patients to indicate the presence of carotid plaques or the possibility of formation of carotid plaques.
Altogether, I recommend the manuscript for publication.
Author Response
Thank you for giving us the opportunity to submit a revised draft of our manuscript entitled: "Association of circulating miR-145-5p and miR-let7c and atherosclerotic plaques in hypertensive patients.
We really appreciate the time and effort you and the reviewers have devoted to providing your valuable comments on our manuscript. We are grateful to the reviewers for their insightful comments on this manuscript and feel they have made it significantly stronger. We were able to incorporate changes to reflect suggestions provided by reviewers. We highlight the changes in the manuscript. Below, we provide a point-by-point response to the reviewers' comments and concerns.
Response Point 1: Thanks for pointing this out. As suggested by the reviewer, the sentence in the summary was modified as follows: In summary, our data demonstrate that hypertensive patients with carotid plaque have increased expression of miR-145-5p and miR-let7c, suggesting that these miRNAs can be used as potential biomarkers of carotid atherosclerosis in patients with arterial hypertension.
Reviewer 2 Report
The manuscript by Minin et al. is a cross-sectional study on the association between serum expression of miR-145-5p, miR-let7c, miR-92a, miR-30a and miR-451, carotid plaques, and intima-media thickness (cIMT) of 177 patients with hypertension. The paper is of interest and my comments are mostly minor.
Introduction
- Line 36: please rephrase the following “As such are predictors for the risk factor for cardiovascular events”
Methods
- Line 79 please change analyzes to analysis
- Line 80, please specify if the experienced physician who performed the carotid ultrasound was the same for all patients or not. Also, if he/she is among the Authors of the manuscript, please specify the initials
- Line 84, please specify who performed the analysis with ImageJ. Also, why the Authors have used ImageJ for cIMT measurement instead of direct measurement during the ultrasound exam?
- Line 119, please use Chi-Square instead of x2
Discussion
- Line 204: Please specify what the Authors mean with “negative feedback mechanism”
- I would add a table/figure to summarize the proposed biological effects of miR-145-5p and miR-let7c in hypertensive patients and atherosclerotic plaques
Author Response
Thank you for giving us the opportunity to submit a revised draft of our manuscript entitled: "Association of circulating miR-145-5p and miR-let7c and atherosclerotic plaques in hypertensive patients. We really appreciate the time and effort you and the reviewers have devoted to providing your valuable comments on our manuscript. We are grateful to the reviewers for their insightful comments on this manuscript and feel they have made it significantly stronger. We were able to incorporate changes to reflect suggestions provided by reviewers. We highlight the changes in the manuscript. Below, we provide a point-by-point response to the reviewers' comments and concerns.
Point 1: Introduction: Line 36: as such are predictors for the risk factor for cardiovascular events
Response Point 1: Thank you for clarifying this: We rephrased the sentence as suggested to: In the primary prevention setting, the presence of carotid atherosclerotic plaques and increased carotid intima media thickness (cIMT) are considered in the definition of subclinical atherosclerosis and, as such, are predictors of cardiovascular events.
Point 2: Methods: Line 79-
Response Point 2: Thanks for the correction. The word “Analyzes“ was corrected.
Point 3: Methods: Line 80-
Response Point 3: Thank you for the opportunity to clarify this issue: Carotid ultrasound of all patients was performed by the same physician. He is co-author of the manuscript, and the initials have been included in the text.
Point 4: Methods: Line 84-
Response Point 4: Thank you for the opportunity to clarify these issues:
1) We modified the text to: "For each image, five measurements of far-wall cIMT were manually performed by a physician (L.F.R.S.C-R) in plaque-free areas using the ImageJ software (NIH, Maryland, USA) and the…..".
2) Our protocol was originally designed to evaluate not only cIMT but also carotid intima thickness (cIT) and carotid media thickness (cMT) separately, as previously described in references 16 and 17 (Sardeli et al. Atherosclerosis. 2017; 261:169-171; Carvalho-Romano et al. Atherosclerosis. 2020; 310:109-110). The evaluation of cMT and cIT cannot be performed by the standard ultrasound software and was only achieved by using the ImageJ software. Therefore, to keep consistency, all carotid wall thickness measures (cIMT as well as cIT and cMT) were performed using the ImageJ software. Given that the focus of the current study was solely on carotid plaques and cIMT, data on cIT and cMT were not provided.
Point 5: Methods: Line 119-
Response Point 5: Thanks for pointing this out. The X2 symbol has been replaced by the word Chi-Square in the text.
Point 6: Discussion: Line 204- Please specify what the authors mean with “negative feedback mechanism”
Response Point 6: Thank you for the opportunity to clarify this issue. We rephrased the sentence to: “First, increases in mir-145 expression may be part of a negative feedback mechanism that impedes plaque progression, as suggested by Knoka et al. In this case, the increase in miR-145 expression in the vulnerable plaque would prevent its further destabilization. However, we do not have data on miRNA expression in atherosclerotic plaque to support this hypothesis.”
I would add a Table / figure to summarize the proposed biological effects of miR-145-5p and miR-let7c in hypertensive patients and atherosclerotic plaques.
Thanks for the suggestion. I fear that for reasons of space, Biomolecules does not allow another figure, especially with a summary of the study. Anyway, we made a figure summarizing our results and included it in the supplementary material and left the decision to use it up to the Biomolecules reviewers and editor.
Reviewer 3 Report
The importance of miRNAs as markers of atherosclerotic burden among hypertensive patients is still under investigation. The authors were aimed at evaluating the serum expression of miR-145-5p, miR-50 let7c, miR-92a, miR-30a and miR-451, and correlating them with carotid plaques and cIMT 51 in hypertensive patients. It was suggested that both miR-145-5p and miR-let7c might be involved in atherogenesis among hypertensive subjects.
The authors submitted well-designed and ethically substantiated investigation, where written consent was obtained from all participants. The study population was well defined with clear inclusion and exclusion criteria. The methods used in this investigation were well chosen and adequately applied.
The obtained results were scientifically challenged to previously obtained findings of other authors, with suggested clinical significance and possible application appropriately underlined.
The limitations were appropriately disclosed.
Author Response
Thank you for giving us the opportunity to submit a revised draft of our manuscript entitled: "Association of circulating miR-145-5p and miR-let7c and atherosclerotic plaques in hypertensive patients.
We really appreciate the time and effort you and the reviewers have devoted to providing your valuable comments on our manuscript. We are grateful to the reviewers for their insightful comments on this manuscript and feel they have made it significantly stronger. We were able to incorporate changes to reflect suggestions provided by reviewers. We highlight the changes in the manuscript. Below, we provide a point-by-point response to the reviewers' comments and concerns.